# Occurrence and Sources of Polycyclic Aromatic Hydrocarbons and Factors Influencing Their Accumulation in Surface Sediment of a Deep-Sea Depression, Namely, the Tatar Trough (Tatar Strait, the Sea of Japan)

Yuliya Koudryashova [1],*, Tatiana Chizhova [1], Pavel Zadorozhny [2], Anna Ponomareva [1] and Alena Eskova [1]

[1]  Pacific Oceanological Institute of Far Eastern Branch, Russian Academy of Sciences, 690041 Vladivostok, Russia; chizhova@poi.dvo.ru (T.C.); ponomareva.al@poi.dvo.ru (A.P.); alena-esya@mail.ru (A.E.)

[2]  Institute of Chemistry of Far Eastern Branch, Russian Academy of Sciences, 690022 Vladivostok, Russia; zadorozhny@mail.ru

*   Correspondence: koudryashova@poi.dvo.ru; Tel.: +7-42311410

**Abstract:** The concentrations of 14 polycyclic aromatic hydrocarbons (PAHs) in the sediment of the Tatar Trough were studied. Despite the increase in PAH concentrations over recent decades, which is likely the result of the handling and transportation of fossil fuels, PAH levels and ecological risk were found to be low. The spatial pattern revealed that higher PAH concentrations were mainly in the deeper water sites, suggesting that trough slope failure transported the PAHs to the deeper part of the basin. There was no correlation between the PAHs and grain size or the PAHs and organic carbon that is related to the PAH input from a variety of sources and the heterogeneity of organic matter. The PAH composition, isomer ratio, and PCA identified two areas with different PAH sources. The most northern part of the Tatar Trough received petrogenic PAHs that are probably transported downslope from the northern Tatar Strait where fossil fuels are handled in some ports. Another trough part was polluted by the PAHs from the combustion of coal and biomass and the exhaust of marine vehicles. The minor presence of genes responsible for aerobic PAH destruction can be explained by the anaerobic degradation of PAHs or the spontaneous creation of favorable conditions that promote bacterial PAH oxidation.

**Keywords:** PAHs; surficial sediments; grain size analysis; microbial degradation; turbidity currents; the Tatar Trough (Sea of Japan)

## 1. Introduction

Polycyclic aromatic hydrocarbons (PAHs) are a group of persistent organic pollutants consisting of two or more condensed rings. These compounds have a harmful effect on living organisms; in particular, PAHs are carcinogenic and cause developmental and metabolic disorders in marine biota [1], which can negatively impact marine biodiversity and marine ecosystem sustainability.

In the marine environment, as PAHs are hydrophobic, they tend to sorb onto the settling organic particles, thereby entering the bottom sediment [2–4]. Sedimentary PAHs are subject to environmental processes, such as burial [5,6], resuspension and re-entry into the water column [7], or destruction by benthic microorganisms [8]. PAHs are formed by the combustion of any organic material, particularly crude oil/oil products and coal, which leads to contamination of the bottom sediment, especially in the areas around industrial centers, as well as marine areas with active shipping [9–11]. Furthermore, PAH sources in marine water include losses during the transportation of fossil fuels (crude oil and coals), as PAHs are a part of them, wastewater discharges, and seeps of oil and natural gas [12–15].

Because of their environmental resistance, PAHs can be transported over long distances through atmospheric and oceanic currents, eventually entering the remote and deep ocean interiors [16–18]. Submarine depressions function as collectors of persistent organic pollutants. The concentration of polychlorinated biphenyls (PCBs) in the sediment of the southern Mariana Trench was found to be increased compared to that in shallower marine areas [19]. Moreover, findings on the sediment in the Atacama Trench (Pacific Ocean) demonstrated that in the hadal part of the trench, deposition dynamics and carbon turnover influenced the concentration of PCBs, which increased as organic carbon content decreased [20].

The Tatar Trough occupies the central and southern part of the Tatar Strait, located in the northwestern Pacific, between the Eurasian continent and the island of Sakhalin, and the northernmost portion of the Sea of Japan as well. The depth of the trough gradually increases to the southwest from 500 to 2500 m.

In the bottom sediment of the Tatar Trough, PAHs can originate from both natural and anthropogenic sources. The former is related to oil or gas leaks, as the geological structure of the Tatar Trough and the increased concentration of the dissolved methane in the water column indicate natural gas and oil reserves [21,22]. Anthropogenic PAHs are the result of shipping since the Tatar Strait is an area of sufficiently intense traffic between the Sea of Okhotsk and the Sea of Japan and the mouth of the Amur River flowing into the strait as well as between the mainland and Sakhalin Island. Thus, ship-induced wastes (e.g., bilge water discharge) and burning fuel for energy are one of the major PAH contributors to the strait water. Furthermore, several ports with large oil and coal terminals, including those receiving oil through a pipeline that runs at the bottom of the Tatar Strait, are located on the west coast of the Tatar Strait. The cargo turnover at the Vanino Port over the past two decades has increased by five times, and in 2022, the port handled 2 and 30 million tons of crude oil and coal, respectively. In 2006, the De Kastri Port, which is one of the largest oil terminals in the Russian Far East, began operating, and its cargo turnover amounted to 5 million tons in 2022. The breaching of regulations during fossil fuel transshipment and storage leads to the ongoing release of pollution. Annual surveys of petroleum products in the water and the sediment of the coastal shelf zone show that the maximum permissible concentration has been exceeded several times [23]. Although the ports are located north of the Tatar Trough, the hydrological dynamics encourage the spread of polluted waters southward to the trough realm.

Electricity generation and heating on Sakhalin Island and adjacent areas to the Tatar Strait are largely coal-fired, which is another significant emitter of PAHs. It is worth adding that Moneron Marine Nature Park is located on the east slope of the Tatar Trough, and uncontrolled PAH pollution can negatively affect the health of the marine park. Therefore, knowledge of PAH dynamics is required to guide future actions and ensure the ongoing protection of the nature park and marine area surrounding it.

This study aimed to (i) determine the PAH concentrations and their ecological and health risk assessment, (ii) identify PAH sources, (iii) consider factors controlling PAH distribution, and (iv) assess the oxidative degradation potential of the sediment in the Tatar Trough.

## 2. Materials and Methods

### 2.1. Sampling of the Sediments

Marine sediments were collected from the Tatar Trough (the Tatar Strait, the Sea of Japan) aboard the R/V Akademik M.A. Lavrentyev during the May–June 2019 cruise. Figure 1 reports on the study area and location of the cruise stations. The bottom depths at the sampling stations are given in Table 1. The sediment sampling was performed by a box corer, and the PAH survey of the upper 2 cm layer was taken. The samples were frozen immediately and stored at −20 °C.

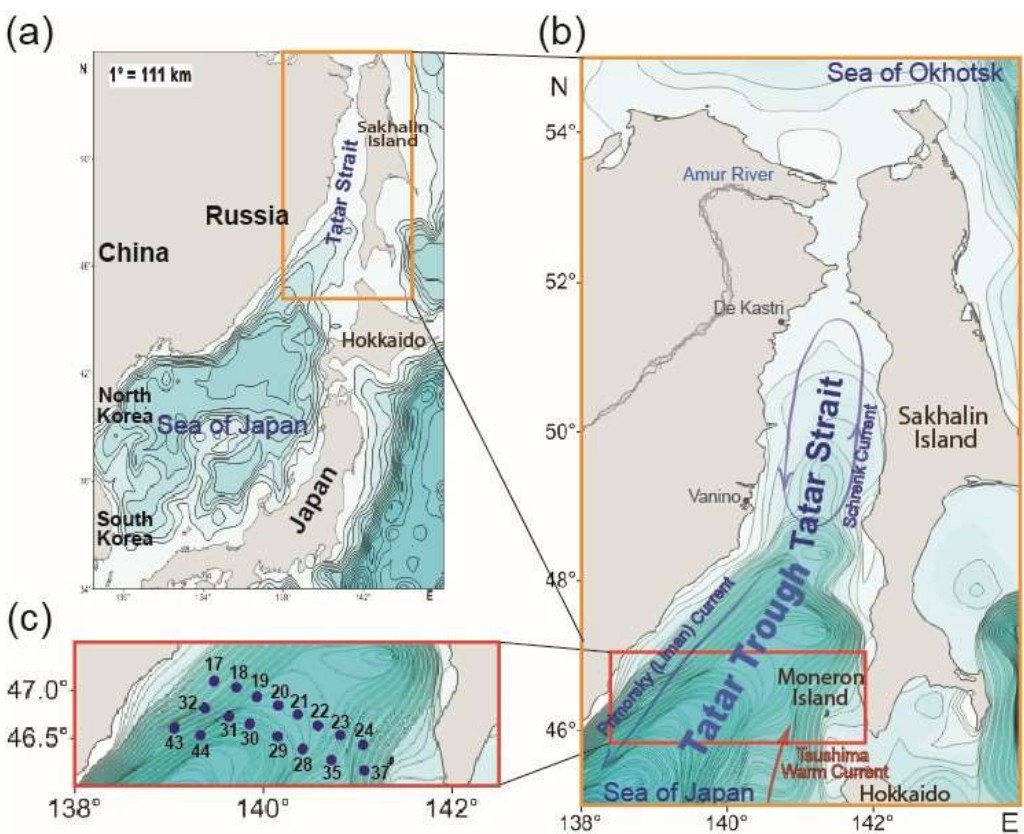

**Figure 1.** (**a**) Map of Tatar Strait relative to the Sea of Japan and Sakhalin Island. (**b**) Map of Tatar Strait with location of Tatar Trough, the main ports, and principal surface currents. (**c**) Study area and station numbers.

**Table 1.** Depths (m) in the sampling stations.

| Station No | 17 | 18 | 19 | 20 | 21 | 22 | 23 | 24 | 28 |
|---|---|---|---|---|---|---|---|---|---|
| Depth, m | 177 | 463 | 796 | 1046 | 1252 | 1412 | 1390 | 804 | 1420 |
| Station No | 29 | 30 | 31 | 32 | 35 | 37 | 43 | 44 | |
| Depth, m | 1290 | 1176 | 1022 | 726 | 1492 | 756 | 780 | 1289 | |

*2.2. Analyses of the Grain-Size and the Organic Carbon*

Upon return to the laboratory, all samples were freeze-dried. The grain size composition was analyzed using a laser particle size analyzer (Analysette 22 NanoTec plus, Fritsch, Germany, Idar-Oberstein), as described in [24]. Classification of the fraction sizes was used as <4 μm (clay), 4–63 μm (silt), and >63 μm (sand), according to the Shepard's scale [25] modified by Schlee [26].

The organic carbon (TOC) was carried out on a TOC-V$_{CPN}$ analyzer with solid sample module SSM-5000A (Shimadzu, Japan) by high-temperature catalytic oxidation, followed by the determination of the concentration of the $CO_2$ by IR gas analysis.

*2.3. PAH's Sample Pretreatment and Analysis*

The lyophilized sediments were passed through a sieve with an aperture of 2 mm before being ball-milled in an agate mortar (Fritsch Pulverisette, Idar-Oberstein, Germany, 30 min). Extraction of PAHs was accomplished by ultrasonication (30 min, twice) of dry milled sediment (5 g) diluted with a mixture of reagent grade dichloromethane (99.9%, Panreac, Barcelona, Spain) and methanol (≥99.9%, Merck, Darmstadt, Germany) (9: 1, *v/v*). The solvents were removed by rotary evaporation, and finally, the residue was evaporated

to dryness under a gentle stream of nitrogen. After reconstituted extracts in acetonitrile, PAHs were characterized by HPLC-fluorescence analysis.

The analysis of sediment extracts was performed on the HPLC system Shimadzu LC-20A equipped with an RF-10Axl detector and Shimpack VP-ODS column (250 mm × 2 mm), according to [27]. PAH analysis was conducted via the external standard method by using mixtures of State standards (PAH acetonitrile solution GSO 10130-2012, Ecroschim, Sankt-Petersburg, Russia). PAH concentration was determined from the calibration curve plotted for each PAH. Then, 14 USEPA priority-listed PAH were quantified: naphthalene (Nap); acenaphthene (Ace); fluorene (Fle); phenanthrene (Phe); anthracene (Ant); fluoranthene (Flu); pyrene (Pyr); benz[*a*]anthracene (BaA); chrysene (Chr); benzo[*b*]fluoranthene (BbF); benzo[*k*]fluoranthene (BkF); benzo[*a*]pyrene (BaP); dibenz[*a,h*]anthracene (DBA); and benzo[*g,h,i*]perylene (BPe). An acenaphthylene did not fluoresce and was, thus, also excluded from this analysis. Since two analytes, Ace and Fle, were not chromatographically separated, they were quantified as a sum.

Recovery Test

Less-polluted sediments taken from the previous cruise in the Arctic region were selected for the recovery study. First, about 5 g of dry-milled sediments was spiked with 1 mL of a 14 PAH solution. A total of 3 solutions containing 14 PAH in the different concentrations (Table 2), which were in the range of the concentrations used to obtain the calibration curve, were used for spiking. The samples were dried overnight in darkness. The PAHs from the samples were eluted using an extraction procedure, as described in Section 2.3. Extraction was carried out in four replicates. Analyses of PAHs were performed by HPLC. The mean recovery rates obtained for each individual PAH are presented in Table 3. The recoveries were used to correct analyte concentrations.

**Table 2.** Solutions (C1, C2, C3) with different PAH concentrations (µg/mL) used for recovery test.

|  | Nap | Ace | Fle | Phe | Ant | Flu | Pyr | BaA | Chr | BbF | BkF | BaP | DBA | BgPe |
|---|---|---|---|---|---|---|---|---|---|---|---|---|---|---|
| C1 | 0.015 | 0.004 | 0.004 | 0.004 | 0.0008 | 0.015 | 0.015 | 0.004 | 0.002 | 0.004 | 0.0008 | 0.0008 | 0.004 | 0.004 |
| C2 | 0.06 | 0.016 | 0.016 | 0.016 | 0.0032 | 0.06 | 0.06 | 0.016 | 0.008 | 0.016 | 0.0032 | 0.0032 | 0.016 | 0.016 |
| C3 | 0.15 | 0.04 | 0.04 | 0.04 | 0.008 | 0.15 | 0.15 | 0.04 | 0.02 | 0.04 | 0.008 | 0.008 | 0.04 | 0.04 |

**Table 3.** Recoveries (%) of PAHs in sediments.

|  | Nap | Ace + Fle | Phe | Ant | Flu | Pyr | BaA | Chr | BbF | BkF | BaP | DBA | BgPe |
|---|---|---|---|---|---|---|---|---|---|---|---|---|---|
| Recovery ± SD, % | 69.7 ± 9.5 | 75.2 ± 10 | 84.8 ± 10 | 71.4 ± 4.5 | 92.2 ± 7.5 | 72.4 ± 6.5 | 77.3 ± 2.2 | 75.7 ± 0.5 | 73.3 ± 2.1 | 73.0 ± 6.0 | 59.09 ± 9.1 | 74.7 ± 3.3 | 65.5 ± 5.0 |

Limits of detection (LOD) and quantification (LOQ) were evaluated as the lowest concentrations of analytes having clear detectable peaks with signal-to-noise ratios (S/N) of 3 and 10, respectively. The LOD ranged between 0.01 pg/injection (Chr) and 5 pg/injection (Nap), and the LOQ varied between 0.04 pg/injection (Chr) and 17.5 pg/injection (Nap). Three laboratory blanks were processed in parallel with the samples during the analytical procedure. The blanks were found to be polluted with Nap, Ace + Fle, Phe, and Pyr, and their concentrations higher than the LOQs were subtracted from the concentrations in the samples.

### 2.4. Aerobic Hydrocarbon Degradation Genes

The total bacterial DNA was isolated from the sediment accordingly [28], with modification. Briefly, 1 g of the sediment was placed in 1 mL of TE buffer (10 mM Tris-HCl pH 8.0, 1 mM EDTA). Then, 20% SDS solution was added to a final concentration of 1–2%. It was thoroughly mixed for 5 min and incubated for 30 min at a temperature of 37 °C.

After incubation, equal volumes of a mixture of phenol and chloroform were added and centrifuged for 10 min at 15,000 rpm. The upper phase was transferred to a new test tube, and an equal volume of a mixture of phenol and chloroform at a ratio of 1:1 was again added; this was repeated twice. Next, 10 μL of 5 M NaCl and 50 μL of 96% ethanol were added and placed in a freezer at −40 °C for at least 2 h. After centrifuging for 15 min at 15,000 rpm, the excess alcohol was removed, and the DNA precipitate was air-dried.

The following functional genes were determined: alkBB—1-monooxygenase gene; narAa; and nahA1 specific for α subunit of the naphthalene dioxygenase (NDO), the catalytic component iron-sulfur protein (ISPNAR) (Table 4). PCR analysis was conducted using Amplifier Dtprime-5 (DNA Technology, Moscow, Russia). The results were analyzed using the software supplied with the instrument RealTime PCR v7.9 (Moscow, Russia).

**Table 4.** Oligonucleotide sequences of primers used.

| Target Compound | Gene | Sequence | Reaction Protocol | Reference |
|---|---|---|---|---|
| Indicator of linear hydrocarbon destruction | *alkBB* | 5′-GGTACGGSCAYTTCTACRTCGA-3′; 3′-CGGRTTCGCGTGRTGRT-5′ | Initial denaturation, 5 min at 94 °C; 35 cycles of 30 s at 94 °C; 30 s at 60 °C; 30 s at 72 °C; final elongation, 8 min at 72 °C. | [29] |
| Indicator of PAH destruction | *nahA1* | 5′-GCCAGATGACCAAGAA ATGGAG TTCC–3′5′-GGCATCGGCATAAATATGTTCGGG–3′ | Initial denaturation, 5 min at 94 °C; 35 cycles of denaturing 92 °C for 1 min; annealing 40 °C for 30 s and extension 70 °C; final elongation, 8 min at 72 °C. | [30] |
| Indicator of PAH destruction | *narAa* | 5′-TACCTCGGCGACCTGAAGTTCTA-3′ 5′-AGTTCTCGGCGTCGTCCTGTTCGAA-3′ | Initial denaturing step at 95 °C for 3 min, followed by 40 cycles of 94 °C for 40 s, 55 °C for 30 s, and 72 °C for 60 s. | [31] |

### 2.5. Ecological Risk Assessment

The presence of PAHs in sediments may pose a threat to human health. In order to evaluate the relative toxicity of sediments, the toxic equivalence quotient (TEQ) and the mutagenic equivalence quotient (MEQ) were calculated based on the Equations (1) and (2):

$$TEQ_{PAH} = \Sigma C_{PAHi} \times TEF_{PAHi} \tag{1}$$

$$MEQ = \Sigma C_{PAHi} \times MEF_{PAHi} \tag{2}$$

where $C_{PAHi}$ represents the concentrations of each PAH congener, and $TEF_{PAHi}$ and $MEF_{PAHi}$ represent toxic and mutagenic equivalent factors for each PAH congener, respectively.

### 2.6. Data Analysis

For the analysis of descriptive statistics, the Pearson and Spearman correlation tests and principal component analysis (PCA) were performed with STATISTICA Software (Version 10, StatSoft Inc., Tulsa, OK, USA). The Shapiro–Wilk test was used to decide whether the distribution of our data was normal or not. The Pearson correlation test was used when variables were normally distributed; otherwise, Spearman's rank correlation test was used.

PCA is a multivariate statistical method for examining factors to reveal relationships and patterns within datasets. Prior to analysis, the original dataset of PAH concentrations was standardized by scaling the values to the mean and standard deviation. Data submitted to the analysis were arranged in a matrix composed of 14 variables (PAH compounds) and 17 cases (the number of sample sites). A varimax rotation was performed to simplify the interpretation results. The number of factors extracted was dictated by eigenvalues being greater than 1. The results of the PCA are presented by loading and score plots.

## 3. Results and Discussion
### 3.1. PAH Level and Comparison with Background

The concentrations of the total 14PAH in the surface sediment varied from 33.4 ng/g to 265 ng/g with an average value of 175 ng/g (Supplementary Materials Table S1). Compar-

ing the PAH concentrations obtained from data on PAHs in the surface sediments of other water bodies around the world (Supplementary Materials Table S2) and data from [32] allowed us to estimate that the total PAH level was generally lower than average on the worldwide scale. However, Phe, BaA, and Chr reached concentration values higher than the worldwide average.

Regarding the temporal trend, in 1993–1994, Nemirovskaya, when studying sediment contamination on the Sakhalin shelf, found that the average PAH content was 1.8 ng/g; this concentration was determined as the background [33]. In 2002, research on sedimentary PAHs revealed that the PAH concentration range varied from 3 to 59 ng/g [34]. It is clear that, from the beginning of the 1990s to the present moment, PAH concentrations have increased tenfold. The upward trend in the PAHs is certainly a consequence of a growing anthropogenic load on the marine area around Sakhalin Island, as oil production started at large fields on the northeastern shelf of Sakhalin in the late 1990s, and coal production in the region was developing and growing, reaching a new all-time high. Obviously, the volume of traffic has increased significantly due to fossil fuel carriers, resulting in enhanced PAH input from the vessels, with atmospheric emission and leakage going directly into the Tatar Trough environment.

Moreover, PAHs can be transported to the Tatar Trough from the northern Tatar Strait area, where ports with large oil and coal terminals are located. The results of a PAH survey in the Amur River estuary and adjacent marine area showed that the $\sum$12PAH concentration in the marine particulate matter was 1146 ng/g, which was twice as high as compared to estuarine particulate matter [35]. This finding indicates that the seawater in the northern Tatar Strait receives greater PAH pollution compared to the estuarine water, which is highly likely to be a result of shipping and seaport operations. The Schrenk and Primorsky currents can transfer this highly polluted water from the north to the south (Figure 1), contributing PAHs to the Tatar Trough. In addition, underwater turbidity currents may relocate the PAHs resuspended from the sediment of the northern Tatar Strait across the strait into the Tatar Trough, which will be discussed below.

*3.2. Ecological Risk Assessment*

The Sediment Quality Guidelines (SQGs) offer a reference point for assessing the potential risk of PAH contamination to aquatic life. The effects range—low (ERL) and effects range—median (ERM) relate to threshold concentrations of individual and total PAHs, determining the probability of adverse biological effects [36]. The ERL corresponds to the 10th percentile of the effects data, a concentration below which the effects are infrequently observed, and higher concentrations of the ERM correspond to the 50th percentile of the effect concentrations. All concentration values of individual and total PAHs obtained were significantly lower than the ERL. This result means that toxic effects caused by PAHs on aquatic organisms rarely occur.

BaP has been listed by The International Agency for Research on Cancer (IARC) of the World Health Organization as a class I human carcinogen [37], and the BaP concentration is used as an indicator of environmental risk. The USEPA has established a permissible concentration threshold of 200 ng/g of BaP in the sediment [38]. The concentration of BaP in the Tatar Trough ranged between 0.34 and 17.7 ng/g, which is much lower compared to the threshold.

Furthermore, the USEPA has included the following PAHs in Group B2 (probable human carcinogens): BaA; Chr; BbF; BkF; and DBA [39]. To evaluate the potential human health risk originating from the sediment pollution by these compounds and BPe, we calculated TEQ and MEQ using Equations (1) and (2) (Section 2.5) since the PAHs have different toxicities. The TEF and MEF values used and the carcinogenic and mutagenic potencies of the PAHs in the Tatar Trough sediment are provided in Table 5. The TEQ describes the ability of PAH congeners relative to BaP to cause tumors and cancer in mammals in vivo and in vitro [40]. The total TEQ values varied from 1.5 (St. 17) to 27.1 (St. 28), with an

average of 10.9 ng TEQ/g. BaA, BaP, and DBA were the main contributors to the total carcinogenic potency, accounting for 85% of the seven PAHs studied.

**Table 5.** TEF and MEF values of carcinogenic PAHs in sediments of the Tatar Trough.

|       | TEF   | MEF   | TEF$_{PAHi}$    | MEF$_{PAHi}$    |
| ----- | ----- | ----- | --------------- | --------------- |
| BaA   | 0.1   | 0.082 | 2.05 ± 0.84     | 1.68 ± 0.69     |
| Chr   | 0.01  | 0.017 | 0.49 ± 0.20     | 0.83 ± 0.33     |
| BbF   | 0.1   | 0.25  | 0.51 ± 0.22     | 1.27 ± 0.55     |
| BkF   | 0.1   | 0.11  | 0.09 ± 0.09     | 0.10 ± 0.10     |
| BaP   | 1     | 1     | 5.15 ± 6.27     | 0.10 ± 0.10     |
| DBA   | 1     | 0.29  | 2.41 ± 1.16     | 0.70 ± 0.34     |
| BgPe  | 0.01  | 0.19  | 0.20 ± 0.09     | 3.79 ± 1.69     |
| Σ7PAH |       |       | 10.90 ± 8.05    | 13.52 ± 8.92    |

The MEQ is based on the ability of a PAH compound to cause DNA modification in human B-lymphoblastoid cells [41]. Its total value in the seven PAHs was calculated to be from 1.8 (St. 17) to 31.8 (St. 28), and the mean was 13.5 ng MEQ/g. The concentrations of BaA, BaP, and BPe posed the highest mutagenic risk and contributed 75% of seven PAHs. Comparing their average values with that in the sediment of marine areas worldwide, the TEQ and MEQ in the Tatar Trough were lower than in estuaries, i.e., the Tiber River estuary [42], the Mahanadi River Estuary, India [43] or in coastal areas, i.e., off Greece [44], San Diego Bay marinas, California [45], the Persian Gulf [46], as well as in the Bohai Sea and the Yellow Sea [47]. From this comparison and compared to the BaP permissible concentration, it is inferred that the cancer- and mutation-inducing ability of the sediment in the present study is low.

### 3.3. Spatial PAH Distribution

The results of the spatial PAH distribution revealed a heterogeneous pattern. In general, the smallest PAH concentrations were found to be closer to the continental and Sakhalin Island shelves, while most of the highest concentrations were observed at the bottom of the deep-water sites of the Tatar Trough (Figure 2). Even a positive moderate correlation between the total PAH concentration and the sampling depth was observed ($p$ = 0.02, $n$ = 17). This pattern differs from the typical circum-continental PAH distribution when the maximum concentration is usually found closer to the coastline where industrial and human activities are located or near the estuaries of polluted rivers, while toward the open deep sea, the PAH content decreases [48–50]. However, the PAH concentration in the Yellow Sea sediment similarly increased with an increasing distance from the coastline toward the open sea, which is explained by the specific hydrologic dynamics and sediment mode [51]. Previously, it was established in the Tatar Trough that sediment accumulation occurs not on the shelf slope but mostly lower [52]. Accordingly, the high PAH concentration in the deep sites can be attributed to sediment deposition mode in the study area.

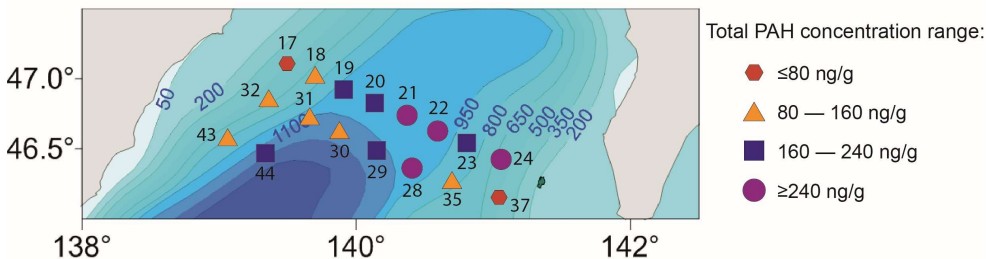

**Figure 2.** Spatial distribution of PAH concentration levels in the sediment of Tatar Trough.

The Passega diagram is a binary plot, with the coarsest 1% of sediments (C) versus the median grain size (M), and it allows for the featuring of the processes of sediment transport and deposition [53]. The CM diagram of the sediment of the Tatar Strait shows a

rectilinear CM pattern, approximately parallel with line C=M (Figure 3), which can be an indicator that the sediment deposition was formed by turbidity currents. These currents are underwater gravity flows triggered by the mobilization of sediments on a shelf slope, and they transfer the sediments from shallow to deep water, so they are capable of transporting organic matter [54] and pollutants, such as plastic, to deep realm [55]. It can be expected that sedimentary particles with PAHs are eroded from the continental shelf by turbidity currents and then resedimented to the bottom of the Tatar Strait. This PAH redistribution can result in PAH abundance in the sediment of deep-water sites, including PAHs from both the vertical flux of biogenic organic particles and shelf sedimentary particles.

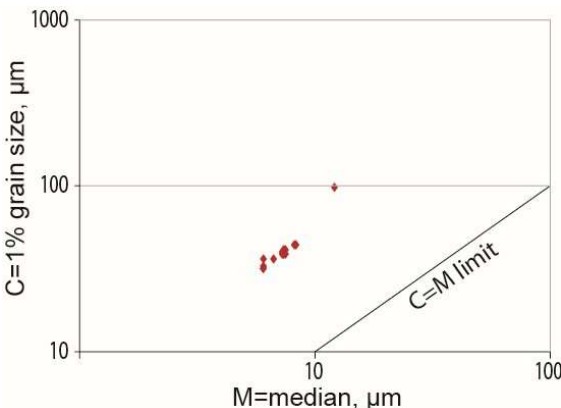

**Figure 3.** CM plot (logarithmic scale) of grain size data.

### 3.4. Grain Size

The sedimentary grain size is one factor affecting the PAH distribution at the bottom because a decrease in the particle diameter of sorbent leads to an increase in its surface area and, accordingly, its sorbent sorption capacity. The relative contribution of each grain fraction in the sediment is presented in Figure 4. According to the classification in [32], most of the sediment sample was clayey silt, and a smaller amount was silt.

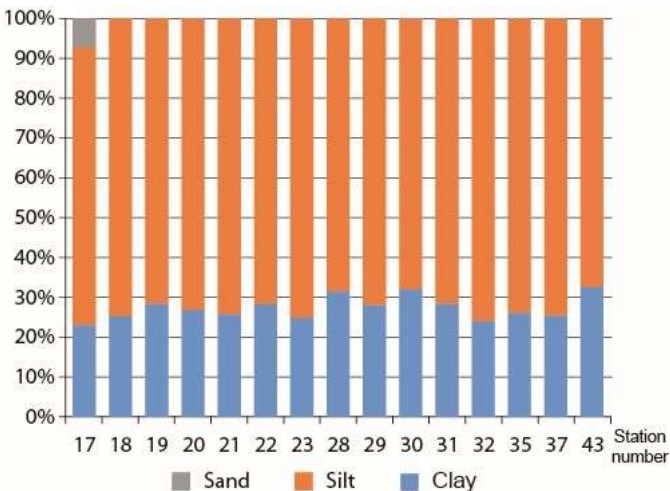

**Figure 4.** Relative abundance of different grain size fractions (clay (<4 μm), silt (4–63 μm), and sand (>63 μm)) in the sediment of Tatar Trough.

No correlation between the total PAH concentration and the relative contribution of the different size fractions was found in the present study. However, in previous studies, a relationship between the PAH concentration and the particle size was observed in the sediment in the northeastern Sakhalin shelf, where the average PAH content in the clay fraction was about three times higher than that in the coarse fraction [34]. A result of the

survey of the marsh and shelf sediment in the northern Gulf of Mexico suggested that the mineral surface area may be important for controlling the PAH distribution when the type of organic matter and the contamination source are similar [49]. It is most likely that until the early 2000s, the number of PAH emission sources increased due to the building of large oil and coal transport hubs and enhanced fossil fuel transport activity. Moreover, PAHs started to enter the sediment in different ways (with oil leaks, vertically settling particles, and lateral turbidity currents), and all of these have led to the total PAH content not depending on the sediment grain size at present.

### 3.5. Organic Carbon

In the sediment of the Tatar Trough, the TOC concentration varied from 4.6 to 22.4 mg/g, with an average value of 18.2 mg/g. The highest concentrations were found closer to the shoreline. Organic matter (OM) is considered another factor controlling the PAH concentration in sediment since PAHs are transported to the bottom when they are sorbed onto organic detritus. The relationship between the PAH content and TOC content has been established in much of the research on coastal and deep marine areas [45,56–59]. However, our study did not reveal the relationship between the PAHs and the organic carbon. The reason may be that the sediment samples of the Tatar Trough did not consist of the same ratio of OM fractions of different origins (terrigenous vs. autochthonous) and aging, showing different affinities for PAHs [60,61]. Furthermore, some studies found that the PAH concentration in the bottom sediment is dictated more by black carbon content rather than TOC due to a more efficient sorption capacity [62]. Thus, the heterogeneity of the organic matrix in the sediment could lead to a disruption of the relationship observed between TOC and the PAHs.

### 3.6. The PAH Sources Identification

#### 3.6.1. Compositional Profile

Since PAHs are released into the environment as a complex mixture, their compositional profile is believed to be source-specific. Nevertheless, the PAH composition can also reflect the influence of factors transforming the initial PAH pattern. The relative individual PAH concentration in the sediment of the Tatar Strait is shown in Figure 5a. The most abundant individual PAHs were Chr, Phe, and BaA. At all locations, HMW PAHs (4–6 ring) prevailed, accounting for an average of 65%; similar dominance has been seen in other studies [45,63–65]. The lack of low molecular weight (LMW) PAHs and the prevalence of high molecular weight (HMW) PAHs in the sediment can be related to their octanol–water coefficient ($k_{ow}$) measuring the hydrophobicity of compounds. The $k_{ow}$-depended distribution of PAHs between the sediment and the water means that the HMW PAHs with higher $k_{ow}$ are absorbed onto sedimentary particles, while less hydrophobic LMW PAHs tend to be in the dissolved form.

Additionally, even though bottom sediment is composed of particles settling from the water column, the relative PAH content in the sediment in this study and in the particulate matter of the surface water in the northern Tatar Strait [67] and the Sea of Japan differed [66]. The particulate matter demonstrated that the proportion of LMW three-ring PAHs was larger than those with higher molecular weight (Figure 5b). Moreover, in the Sea of Japan, the values of $\log k_{oc}$ (particle–water distribution coefficient based on particulate organic carbon content) of light PAHs (Ace, Fle, and Ant) are similar to that of heavy-weighted PAHs. This suggests that the environment of bottom sediment (temperature, salinity, pH, etc.) contributes to the achievement of sediment–water equilibrium for PAHs, in contrast to water column conditions. Apart from their hydrophobicity, benthic microorganisms can influence the removal of more easily degraded LMW PAHs from the sediment.

The clear difference in the PAH compositional profiles between sites was detected among most of the stations and Sts. 18–23, at which the Nap content was detected to be from 5.4 to 6.3% (Figure 5b), which is above average (4.8%). Nemirovskaya, when studying the northeastern Sakhalin shelf in the 2000s, similarly found an elevated Nap

level, assuming that the PAH in the bottom sediment may increase with oil leaking from the sedimentary rock [34]. Moreover, Nap is a natural component of coal tar, with a typical content of up to 11% [68]. As mentioned above, a large coal terminal is situated in the Vanino Port on the mainland coast to the north of the sampling location. Probably, the coal dust deposited in the sediment of the strait zone adjacent to the port could be resuspended by the turbidity currents triggered by the change in the Amur River discharge and water dynamic [69] and, accordingly, moved south through the northern tip of the Tartar Trough into the sediments of the closest stations, namely, Sts. 18–23.

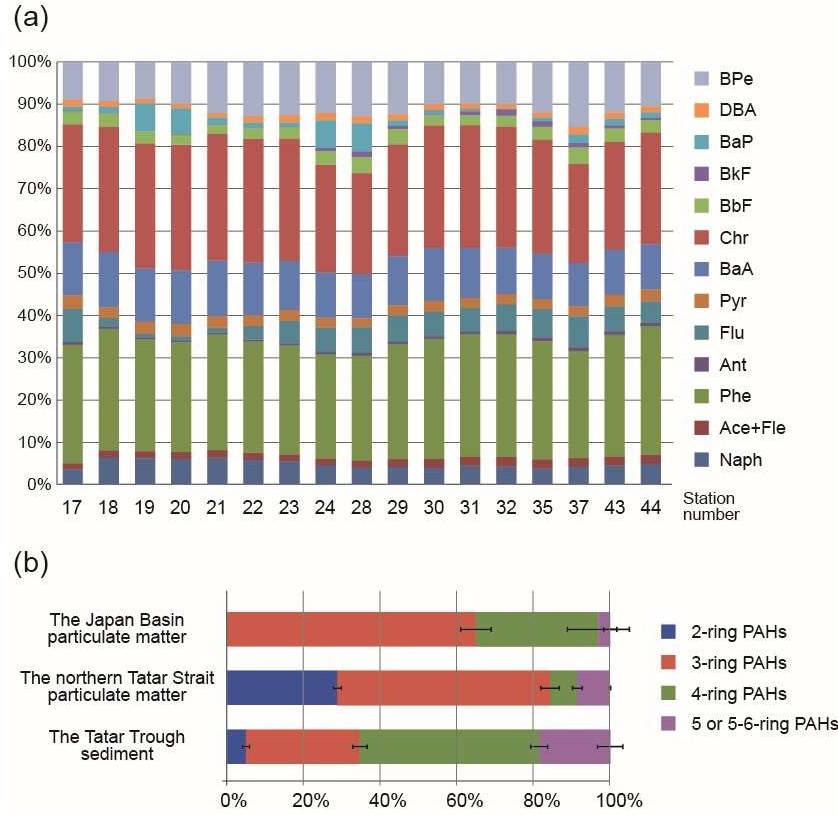

**Figure 5.** (**a**) Percent contribution of individual PAHs (Ace and Fle were quantified as a sum) in surface sediment of Tatar Trough. (**b**) Comparison of relative contribution of 2-, 3-ring PAHs, 4-ring PAHs, and 5- and 6-ring PAHs (mean ± SD) in particulate matter in the water column of the Japan Basin (based on data from [66]), particulate matter in surface water of northern Tatar Strait (based on data from [67]), and sediment of Tatar Trough.

The largest relative concentration values of five–six-ring PAHs were found at Sts. 24, 28, and 37. These stations are located off the fairly populated southern Sakhalin Island and on active marine traffic routes. Thus, the five–six-ring PAHs can be a result of combustion in various motor vehicles and fuels for heating. Additionally, it is known that the areas of these stations are under the influence of the Tsushima Warm Current [70] flowing from the East China Sea and carrying continental shelf water polluted with PAHs. The main sources of PAHs in the water of the Tsushima Warm Current were determined to be combustion products [71]. Moreover, when considering individual five–six-ring PAHs, the samples of St. 37 contained the highest percentage contribution of BbF and BPe, which is the same pattern found for particulate PAHs in the water of the Tsushima Warm Current. This correspondence supports the PAH input by the Tsushima Warm Current in the southern Tatar Strait.

### 3.6.2. PAH Isomer Ratio

The PAH isomer ratio method is commonly used to determine the PAH origin ([72] and references therein) since it is believed that PAHs emitted from one source are in certain concentration ratios. Moreover, PAH isomers, as they have similar physicochemical properties, undergo similar changes during transport in the environment. However, the diagnostic ratios method should be used with caution and interpreted in combination with other methods because it is often difficult to differentiate among some sources.

Figure 6 shows the results obtained for the following ratios: Ant/(Ant + Phe); BaA/(BaA + Chr); Flu/(Flu + Pyr); and BaP/BPe. The first two ratios revealed a stable narrow range of values for all stations (Figure 6a). It should be noted that some research has found that for sediment samples, the Ant/(Ant + Phe) better identifies petrogenic PAH sources, while the BaA/(BaA + Chr) is a good indicator of pyrolytic pollution emission sources [72]. The Ant/(Ant + Phe) values ranged from 0.2 to 0.3, indicating petrogenic PAH. BaA/(BaA + Chr) was in the range of 0.28–0.31, which means a mixture of petrogenic and pyrogenic sources of PAHs.

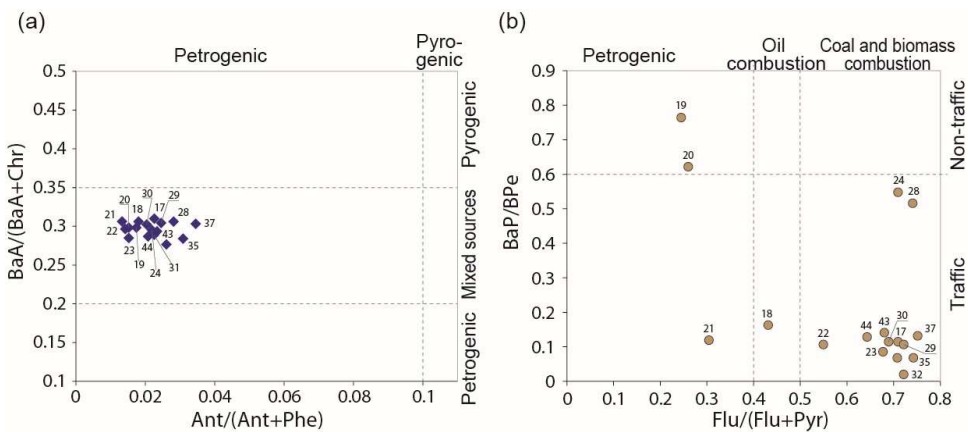

**Figure 6.** Plot showing PAH isomer ratio: (**a**) Ant/(Ant + Phe) vs. BaA/(BaA + Chr); (**b**) Flu/(Flu + Pyr) vs. BaP/BPe. Squares and circles denote the sampling stations.

The Flu/(Flu + Pyr) determined that PAHs were of pyrogenic origin derived predominantly from coal and biomass combustion at most stations, except the samples from Sts. 19–21 (Figure 6b). Pyrogenic PAHs are obviously a consequence of coal combustion, which provides most of the energy for domestic and central heating to Sakhalin Island and the continental region around the study area. In addition, forest fires in the Russian Far East and Siberia, from where PAHs can be transferred with atmospheric aerosols [73], can cause the occurrence of pyrogenic PAHs in the sediment samples. Sts. 19–21 featured PAHs of petrogenic origin. This finding corresponds to the increased Nap concentrations observed at these sites, which confirms that the PAHs were from crude oil/coal. However, it is still difficult to accurately distinguish whether natural or anthropogenic causes led to the fossil fuel sedimentary contamination.

The BaP/BPe ratio is a tool to distinguish between traffic and non-traffic PAH sources [74]. The ratio values showed that the majority of stations received PAHs from the exhaust of vehicles, while the samples from Sts. 19 and 20 contained PAHs from non-traffic emissions, and the result of St. 24 was close to this as well (Figure 6b). The results of the BaP/BPe ratio taken with the Flu/(Flu + Pyr) ratio and the PAH compositional profiles confirm that the area around Sts. 19–21 has different PAH sources from the other sampling localities. As mentioned above, this difference can be related to the intrusion of turbidity currents delivering sedimentary PAHs that resulted from ports handling fossil fuels to the trough part where Sts. 19–21 were located, while the PAHs at the other stations originated predominantly from various types of combustion and could enter the bottom sediment by both vertical and lateral pathways.

### 3.6.3. Principal Component Analysis

PCA is often used to differentiate PAH sources, as it simplifies the interpretation of highly variable data by reducing their set and grouping variables with similar characteristics into factors while still explaining most of the variance. It is worth noting that the factors can be explained by not only the potential PAH origin but other causes that influence the PAH pattern in the environment as well.

The first factor corresponded to 42% of the total variance and was determined by the largest positive values of loadings of Chr, Nap, and BaA and negative values of Flu and Phe (Figure 7a). It is difficult to attribute the grouping of compounds to whether these PAHs are of pyrogenic or petrogenic origin here since both of the PAH groups similarly include both three- and four-ring PAHs. However, if we take into account that the Flu/(Flu + Pyr) ratio showed the petrogenic origin of the PAHs in the samples of Sts. 19–21 (Figure 6b), where the highest factor loadings of Nap, Chr, and BaA were detected (Figure 7a,b) and where these PAHs were parts of various types of coal [75,76], it can be assumed that this group of PAHs, on the positive side of the axis, is associated with petrogenic sources. As for PAHs loading the negative axis, Phe was found to be the largest PAH released from coal combustion in power plants and boilers ([77] and references therein). Moreover, the boilers use forest biomass and bituminous coal as fuel and release BbF, Phe, DBA, and Flu in the largest quantities. As can be seen, all these PAHs, except DBA, contribute to the negative loading of Factor 1 (Figure 7a). Since the combined use of coal and wood biomass is often used for heating in households in this region of Russia, it can be assumed that this group of PAHs is of pyrogenic origin.

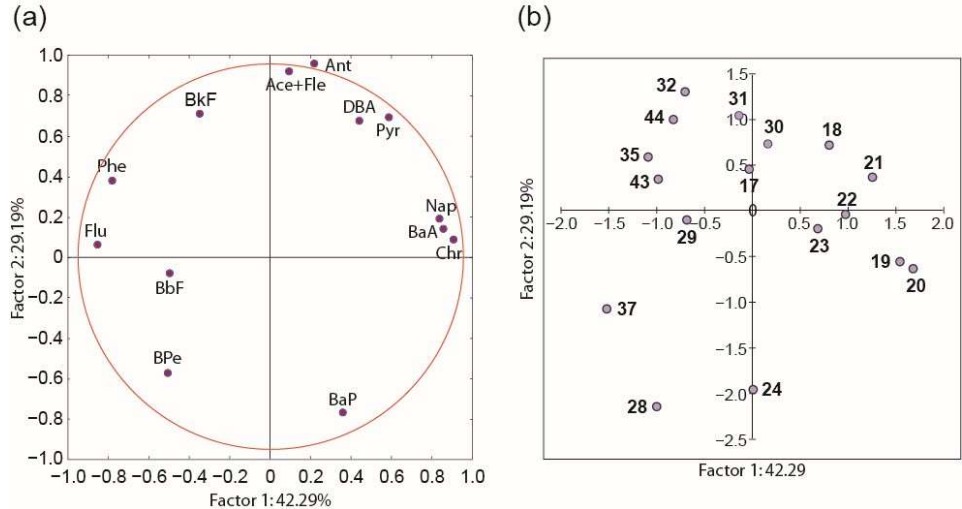

**Figure 7.** Result of principal component analysis (PCA) for PAHs in the sediment of Tatar Trough: (**a**) Load plot; (**b**) Score plot. Circles with numbers denote the sampling stations.

Additionally, as mentioned above, Sts. 19–21, with the highest loads of Chr, Nap, and BaA, differ from the rest in that they receive laterally transported sedimentary PAHs. Hence, it is likely that PAHs with negative loading values came to the bottom sediment with settling organic particles. In other words, Factor 1 divided the PAHs not according to their petrogenic or pyrogenic origin but based on such criteria as the manner of PAH transport.

The second factor was responsible for 29% of the total variance and was highly weighted in Ace + Fle, Ant on the positive axis and in Bap and BPe on the negative side of the axis. These results imply that the PAHs clustered depending on the source; in other words, the grouping of LMW PAHs indicates their petrogenic origin, and that of HMW PAHs indicates a pyrogenic one. However, it is worth considering that along with LMW PAHs, HMW Pyr, BkF, and DBA also contribute to the positive factor load that causes the contradiction that these PAHs are derived from petrogenic sources. Given that LMW PAHs degrade easily in the environment compared to HMW PAHs [78] clustering on the

positive axis side of Ace + Fle, Ant can signify that the vertical particle flux more intensively delivered organic matter to the bottom of the localities of the samples with greater LMW PAH contributions. An increased organic carbon concentration was found, indicating the enhanced particle flux in the water spreading along the mainland [69], where samples with high loads of Ace + Fle, Ant were taken (Sts. 31, 32, and 44) (Figure 7a,b).

On the other hand, unlike the sediments of the sites where a supply of fresh organic matter is sustained, the sediments at St. 24 and St. 28 with the lowest loads of Ace + Fle, Ant (Figure 7a,b), located at a great distance from the coast of Sakhalin Island and the mainland, are probably under weak biological pump activity, and a low sedimentation rate led to the depletion of LMW PAHs with the aging of the organic matter and the accumulation of more refractory HMW PAHs.

### 3.7. The Sediment Degradation Potential

Microbial degradation is believed to be one of the major pathways to purifying PAH-contaminated sediments. In a previous study, PAH-oxidizing bacteria isolated from the coastal water and sediments of the Sea of Japan and the Tatar Strait demonstrated high oxidative activity to decompose PAHs, which was explained by the severe oil pollution of their habitat [79,80]. Aerobic PAH degradation is often catalyzed at the first step by ring-hydroxylating dioxygenases, which incorporate oxygen atoms into the benzene ring [8]. Functional genes encoding subunits of these enzymes are used as indicators of PAH degradation [81,82]; their presence indicates whether PAH destruction occurred during the sampling. We surveyed our sediment samples to find the nahA and narA genes responsible for encoding naphthalene-degrading enzymes found in the genus Pseudomonas of Gram-negative bacteria and Gram-positive Rhodococcus strains, respectively. Both genera are composed of genetically and physiologically diverse bacteria commonly found in various habitats [83–85], and their presence was revealed in the coastal environment of the Sea of Japan and the Tatar Strait [79,80,86].

The results show, in the samples, that there were no nahA genes and only the presence of narA at St. 30. The lack of the genes, implying no aerobic degradation of the PAHs, can be associated with the bacterium preferring to obtain energy from a more available substrate, such as alkanes. This is confirmed by the fact that we found the AlkBB gene, which encodes alkane hydrolases, at all the stations examined. Moreover, PAHs could be transported to the study area with aged OM, making PAHs less available to the biota [87]; this could also provoke bacteria to use more convenient energy sources.

Nevertheless, a strong reduced content of light PAHs in the sediments compared to the water column potentially indicates processes of microbiological destruction. Therefore, it can be assumed that conditions favorable for aerobic PAH decomposition could be created spontaneously, for example, due to seasonal hydrodynamic changes or by bacteria-degraded PAHs anaerobically. It is believed that methanogenic environments, such as crude oil reservoirs and fuel-contaminated sediments, can support syntrophic PAH bacterial metabolism coupled with methane production, which is one anaerobic pathway to PAH destruction. The degradation of some LMW PAHs under methanogenic conditions has been reported in contaminated soils, sewage and petrochemical sludge, and sediments ([88] and references therein). Ponomareva [89] investigated geomicrobiological indicators specific to the gas-hydrate and non-gas-hydrate marine areas in the northern Sea of Japan and the Tatar Strait and revealed the presence of genes of methanogenic microorganisms. These findings and our results suggest that methanogenic anaerobic degradation can be one route to the detoxification of PAH in the Tatar Strait sediment.

In conclusion, this research provides insights into the PAH level in the sediment of the Tatar Trough, areas with occurrences of PAHs of natural and anthropogenic origin. Sediment deposition mode was established to be the factor determining the PAH accumulation. Locations with different PAH origins, pyrogenic or petrogenic, were identified, but it was still difficult to distinguish whether the PAHs were derived from natural or anthropogenic sources. Thus, a detailed study is needed in the future to better understand the pollution

pathways and more accurately estimate the influence of sedimentation rates, degradation processes, and the quality of organic matter on the fate of PAHs.

**Supplementary Materials:** The following supporting information can be downloaded at https://www.mdpi.com/article/10.3390/w15234151/s1: Table S1: The PAH concentrations (ng/g) in sediments of the Tatar Trough; Table S2: Review of PAHs values presented for different areas; References [90–101] are cited in Supplementary Materials.

**Author Contributions:** Conceptualization, Y.K.; methodology, Y.K. and T.C.; validation, T.C., Y.K. and A.P.; formal analysis, T.C. and Y.K.; investigation, T.C., Y.K., A.P., A.E. and P.Z.; resources, T.C. and A.P.; writing—original draft preparation, Y.K., T.C. and A.P.; visualization, Y.K. and T.C.; project administration, T.C., Y.K. and A.P.; funding acquisition, T.C. All authors have read and agreed to the published version of the manuscript.

**Funding:** This work was performed as part of the Russian Federation State Assignment of the V.I. Il'ichev Pacific Oceanological Institute AAAA-A19-119122090009-2 and the Russian Science Foundation (grant no. 23-77-10038).

**Data Availability Statement:** Any datasets not included in this published work are available upon reasonable request from the corresponding author; however, all data produced or analyzed during this investigation are contained in it.

**Acknowledgments:** The authors would like to thank the Far Eastern Center for Structural Research of the Institute of Chemistry FEB RAS for assistance in conducting the HPLC analysis.

**Conflicts of Interest:** The authors declare no conflict of interest.

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
