# Peer review of "Occurrence and Sources of Polycyclic Aromatic Hydrocarbons and Factors Influencing Their Accumulation in Surface Sediment of a Deep-Sea Depression, Namely, the Tatar Trough (Tatar Strait, the Sea of Japan)"

_water, doi:10.3390/w15234151_

Round 1

Reviewer 1 Report

Comments and Suggestions for Authors

The manuscript under the title: (Occurrence and sources of polycyclic aromatic hydrocarbons and factors influencing their accumulation in surface sediment of deep-sea depression namely Tatar Trough (Tatar Strait, the Sea of Japan) showed excellent addition to understanding the geochemistry of PAHs in the studied area. Although the data showed a revere pattern from the normal expected spatial distribution of PAHs, where the minimum PAHs concentrations were found close to the continental shelves near human activities, and the highest concentrations were observed at the bottom of the deep-water sites far from the coastal area, however, the authors (in my opinion), brilliantly managed to put a very acceptable explanation for this reverse situation. The part of microbial degradation of PAHs is very important. The authors were very careful in dealing with the origin sources of PAHs either in the isomeric ratios or in the PCA which is a good point.

I recommend publishing this paper in the Water Journal after considering the following comments:

1-     Keywords: polycyclic aromatic hydrocarbons; PAH; sediments; organic pollution; turbidity currents; organic carbon; grain size; the Tatar Trough; the Sea of Japan

Seems too many keywords. You can make it:

PAHs; surficial sediments; grain size analysis; microbial degradation; turbidity currents; the Tatar Trough (Sea of Japan)

2-     Page 2 line 54: something wrong in the sentence (language)

3-     In the recovery test: would you please mention the concentrations of PAH mixture used for spiking the dry milled sediments? What are the three different concentrations that were used to obtain the calibration curve?

4-     Page 4, line 143: would you please mention the ranges of LOD and LOQ for PAHs.

5-     Page 4, line 157: would you please mention the volume of NaCl and methanol.

6-     In the first paragraph of the discussion (PAH level and comparison background), comparing with other previous studies is very important, but more important to compare with fixed values for individual and total detected PAHs like sediment quality guide liens (SQGs), it will give a more precise comparison, moreover it will give an impression about the risk assessment for living organisms.

7-     Table S2: change (min-max) to (average).

8-     Line 192: please adjust the language.

9-     The correlation between organic carbon and PAHs in your work seems confusing for the reader. It usually is a strong positive correlation. Your explanation classified PAHs in this case to HMW and low or middle PAHs and finally no correlation with total PAHs, while individual low and high MW PAHs give negative correlation. I think this part needs changes in the strategy of discussion.

10- Page 9, line 298; please change (The relative PAH) to (The relative individual PAH).

11- Page 9, line 303: please adjust the language.

12- In the (Compositional profile) section: you attribute the low concentration of LMW PAHs in the sediments only to (more easily degraded by benthic microorganisms that lead to their depletion in the bottom sediment). You can add another important reason which is the relatively high solubility of LMW PAHs over HMW PAHs that allow sediment-water exchange and decrease concentration of LMW PAHs in the sediments.

13- Page 10, line 331: (concentrations of carcinogenic 5-6-ring PAHs)

It will be very helpful to understand this part if you calculate the mutagenic equivalence quotient (MEQ) and toxic equivalence quotient (TEQ) of carcinogenic PAHs.

Comments on the Quality of English Language

A few sentences need minor English language corrections (part of them I mentioned in my comments).

Some commas are also missing in some sentences. 

Author Response

We sincerely thank the reviewer for taking the time to review our manuscript. Your comments helped improve the manuscript, especially the discussion, which became much clearer and more relevant to the results.

Additionally, please pay attention to that the data on PAH concentration was changed. While working on correcting the manuscript, we found that we made a mistake when converting data in solution to dry weight. As a result, the PAH concentrations decreased by 2.5 times and thus the discussion of pollution levels was slightly modified. However, this did not affect the results of the relative PAH contribution (compositional profiles), correlation analysis, PCA and isomeric ratio.

1-     Keywords: polycyclic aromatic hydrocarbons; PAH; sediments; organic pollution; turbidity currents; organic carbon; grain size; the Tatar Trough; the Sea of Japan

Seems too many keywords. You can make it:

PAHs; surficial sediments; grain size analysis; microbial degradation; turbidity currents; the Tatar Trough (Sea of Japan)

The keywords were changed.

2-     Page 2 line 54: something wrong in the sentence (language)

The sentence was checked.

3-     In the recovery test: would you please mention the concentrations of PAH mixture used for spiking the dry milled sediments? What are the three different concentrations that were used to obtain the calibration curve?

The information was added.

4-     Page 4, line 143: would you please mention the ranges of LOD and LOQ for PAHs.

The information was added.

5-     Page 4, line 157: would you please mention the volume of NaCl and methanol.

The information was added.

6-     In the first paragraph of the discussion (PAH level and comparison background), comparing with other previous studies is very important, but more important to compare with fixed values for individual and total detected PAHs like sediment quality guide liens (SQGs), it will give a more precise comparison, moreover it will give an impression about the risk assessment for living organisms.

We created a new section “Ecological Risk Assessment” in the “Results and Discussion” part and included this information.

7-     Table S2: change (min-max) to (average).

Corrected

8-     Line 192: please adjust the language.

The sentence was checked.

9-     The correlation between organic carbon and PAHs in your work seems confusing for the reader. It usually is a strong positive correlation. Your explanation classified PAHs in this case to HMW and low or middle PAHs and finally no correlation with total PAHs, while individual low and high MW PAHs give negative correlation. I think this part needs changes in the strategy of discussion.

We used a different approach to determine the relationship between TOC and PAHs and changed the discussion.

10- Page 9, line 298; please change (The relative PAH) to (The relative individual PAH).

Corrected

11- Page 9, line 303: please adjust the language.

The line was changed.

12- In the (Compositional profile) section: you attribute the low concentration of LMW PAHs in the sediments only to (more easily degraded by benthic microorganisms that lead to their depletion in the bottom sediment). You can add another important reason which is the relatively high solubility of LMW PAHs over HMW PAHs that allow sediment-water exchange and decrease concentration of LMW PAHs in the sediments.

The discussion was expanded.

13- Page 10, line 331: (concentrations of carcinogenic 5-6-ring PAHs)

It will be very helpful to understand this part if you calculate the mutagenic equivalence quotient (MEQ) and toxic equivalence quotient (TEQ) of carcinogenic PAHs.

The information was added.

Reviewer 2 Report

Comments and Suggestions for Authors

Review of the article, Water 2690775

Yuliya Koudryashova, Tatiana Chizhova, Pavel Zadorozhny, Anna Ponomareva, and Alena Eskova

«Occurrence and sources of polycyclic aromatic hydrocarbons and factors influencing their accumulation in surface sediment of deep-sea depression namely Tatar Trough (Tatar Strait, the Sea of Japan)».

The reviewed paper deals with PAH studies in surface bottom sediments sampled in the area of Tatar Strait, the Sea of Japan, waters of which are characterized by presence both of anthropogenic and of natural sources of POPs of this class. Potential natural PAH sources are traced by the geological structure of the Tatar trough and the increased concentration of the dissolved methane in the water column. Anthropogenic PAH sources are intensive navigation between the Sea of Okhotsk and the Sea of Japan and the Amur River waters. Modern large oil and coal terminals on the western coast of Tatar Strait are evidently main PAH sources in this area. Thus, studies and assessment of pollution in Tatar Strait area, the Sea of Japan with PAH is of a practical importance. Studies of surface bottom sediments as of natural accumulating matrix, revealing of PAH accumulation levels and sources, revealing of factors determining PAH distribution and especially assessment of the oxidative degradation potential of the sediment is a founded and valuable scientific task.

It is necessary to pay attention to tasks formulation, in particular: ii) to consider factors controlling the PAH distribution and composition such as grain size and organic carbon, grain size and organic carbon – is this already a response to the established task?

To match the paper manuscript up to Water journal level, authors need to find out the answers to the next questions.

PAH determination in bottom sediments was done only for US EPA priority-listed PAH, this is evidently insufficient for identification of PAH sources, both anthropogenic and natural ones. It is evident that to separate sources, it is necessary to determine methylated derivatives of naphthalene, phenanthrene, chrysene, which are hydrocarbons of raw oil, as well as such important proxy PAH as perylene, benz[e]pyrene, retene. Unfortunately, the authors are constrained by range of use of HPLC method with fluorescence detector, while use of GC-MS method would allow to extent the range of studied hydrocarbons. Probably, the authors have to constrain themselves by assessment of PAH pollution levels using for this US EPA priority-listed PAH and entitling the article according ti this aim.

The authors have collected a considerable amount of experimental material, Table S1, but in the manuscript it is presented very incompletely, e.g., in the clause 3.1., there is only one phrase, line 178-179, further, line 180-206, there is a comparative analysis with worldwide data. However, their own data are not presented: e.g., which groups of PAH congeners are found in bottom sediments and what is their ratio. On the reviewer’s opinion, the chosen form of the manuscript presentation decreases very much the own authors’ contribution into the solution of this scientific problem.

In the clause 3.3 Factors influenced the PAH concentration and composition, a way of search of the presented factors is chosen poorly. The presented in Figure 4b, Figure 5 dependencies are not representative and essentially, they manifest the absence of these dependencies. Analysis of data on PAH content in bottom sediments using, e.g., PERMANOVA approach, would evidently allow to reveal searched factors.

Warnes, G.R. et al. Package “gplots”: Various R Programming Tools for Plotting Data, R Package Version 2.17.0; Science Open: Berlin, Germany, 2015. https://cran.r-project.org/web/packages/gplots/index.html

Grzymala-Busse, J.W., at al. Handling missing attribute values in preterm birth data sets. In Proceedings of the International Workshop on Rough

Germany, 2005; pp. 342–351. https://sci2s.ugr.es/keel/pdf/specific/congreso/grzymala_busse_goodwin05.pdf.

At the same time, sources identification by PAH congeners ratio is presented convincingly, Figure 7. I propose to outline in Figure 7а diagnostical intervals for ratios of BaA/(BaA+Chr) and Ant/(Ant+Phe), like in Figure 7b.

While discussing of a series of priority-listed PAH distribution in bottom sediments, in water column, it is necessary to take into account the constants logKow, according to which, hydrophilic PAH will be concentrated in water column, and hydrophobic, high molecular ones – in bottom sediments, the ratio of the latter ones will depend on a pollutants source.

A very important problem is considered in the clause 3.5, the sediment degradation potential, but the authors did not highlight the following matters. Did a microbial community vary on the sites characterized by the increase of PAH sources intensity compared to former years data, if such ones are available, or to data for relatively clean areas? Which was a fixed response of the ecosystem to the increase of natural objects pollution and which is the own potential of sediments for self-cleaning? I think that it would be interesting for the authors to consider an approach to a study of this problem in

«Anaerobic oxidation of petroleum hydrocarbons ...», Pavlova et al., 2021, Microbial Ecology. https://doi.org/10.1007/s00248-021-01802-y

Minute remarks:

I propose to the authors to change the Figure 2, on which the PAH content levels are presented by circles of different sizes and colors, they present the information unconvincingly. Evidently, it is necessary to present PAH content levels with different forms – circles, triangles, squares and to choose more contrast colors. It would be very useful to mark the depth of sampling.

The authors did not indicate the accuracy of PAH determination, the measurements results (a number of significant figures) do not meet the value of confidence intervals in Table S1 and in the text.

Authors did not indicate measurement units for PAH concentration for limits of detection, quantification and laboratory blanks.

Author Response

We sincerely thank the reviewer for taking the time to review our manuscript. Your comments helped improve the manuscript, especially the discussion, which became much clearer and more relevant to the results.

Additionally, please pay attention to that the data on PAH concentration was changed. While working on correcting the manuscript, we found that we made a mistake when converting data in solution to dry weight. As a result, the PAH concentrations decreased by 2.5 times and thus the discussion of pollution levels was slightly modified. However, this did not affect the results of the relative PAH contribution (compositional profiles), correlation analysis, PCA and isomeric ratio.

The reviewed paper deals with PAH studies in surface bottom sediments sampled in the area of Tatar Strait, the Sea of Japan, waters of which are characterized by presence both of anthropogenic and of natural sources of POPs of this class. Potential natural PAH sources are traced by the geological structure of the Tatar trough and the increased concentration of the dissolved methane in the water column. Anthropogenic PAH sources are intensive navigation between the Sea of Okhotsk and the Sea of Japan and the Amur River waters. Modern large oil and coal terminals on the western coast of Tatar Strait are evidently main PAH sources in this area. Thus, studies and assessment of pollution in Tatar Strait area, the Sea of Japan with PAH is of a practical importance. Studies of surface bottom sediments as of natural accumulating matrix, revealing of PAH accumulation levels and sources, revealing of factors determining PAH distribution and especially assessment of the oxidative degradation potential of the sediment is a founded and valuable scientific task.

It is necessary to pay attention to tasks formulation, in particular: ii) to consider factors controlling the PAH distribution and composition such as grain size and organic carbon, grain size and organic carbon – is this already a response to the established task?

The tasks have been corrected.

To match the paper manuscript up to Water journal level, authors need to find out the answers to the next questions.

PAH determination in bottom sediments was done only for US EPA priority-listed PAH, this is evidently insufficient for identification of PAH sources, both anthropogenic and natural ones. It is evident that to separate sources, it is necessary to determine methylated derivatives of naphthalene, phenanthrene, chrysene, which are hydrocarbons of raw oil, as well as such important proxy PAH as perylene, benz[e]pyrene, retene. Unfortunately, the authors are constrained by range of use of HPLC method with fluorescence detector, while use of GC-MS method would allow to extent the range of studied hydrocarbons. Probably, the authors have to constrain themselves by assessment of PAH pollution levels using for this US EPA priority-listed PAH and entitling the article according ti this aim.

We agree that the use of methyl-substituted PAHs greatly facilitates the separation of petrogenic and pyrogenic sources. Nevertheless, in case of, for example, crude oil leaks from a pipeline or coal dust entering the water (which is likely to occur in our study area), this will not help separate anthropogenic and natural contamination.

The authors have collected a considerable amount of experimental material, Table S1, but in the manuscript it is presented very incompletely, e.g., in the clause 3.1., there is only one phrase, line 178-179, further, line 180-206, there is a comparative analysis with worldwide data. However, their own data are not presented: e.g., which groups of PAH congeners are found in bottom sediments and what is their ratio. On the reviewer’s opinion, the chosen form of the manuscript presentation decreases very much the own authors’ contribution into the solution of this scientific problem.

The information on PAH congeners and their ratio was described in the section 3.6.1. “Compositional profile”.

In the clause 3.3 Factors influenced the PAH concentration and composition, a way of search of the presented factors is chosen poorly. The presented in Figure 4b, Figure 5 dependencies are not representative and essentially, they manifest the absence of these dependencies. Analysis of data on PAH content in bottom sediments using, e.g., PERMANOVA approach, would evidently allow to reveal searched factors.

Warnes, G.R. et al. Package “gplots”: Various R Programming Tools for Plotting Data, R Package Version 2.17.0; Science Open: Berlin, Germany, 2015. https://cran.r-project.org/web/packages/gplots/index.html

Grzymala-Busse, J.W., at al. Handling missing attribute values in preterm birth data sets. In Proceedings of the International Workshop on Rough

Germany, 2005; pp. 342–351. https://sci2s.ugr.es/keel/pdf/specific/congreso/grzymala_busse_goodwin05.pdf.

We thank the reviewer for providing information on the approaches to data analysis. This will be really useful for our research.

As for this work, after verifying whether the data are normally distributed we changed the approach and used additionally Spearman's correlation to find the relationship between the studied parameters. Since the correlation was not established, as a result, the line of the discussion got changed.

At the same time, sources identification by PAH congeners ratio is presented convincingly, Figure 7. I propose to outline in Figure 7а diagnostical intervals for ratios of BaA/(BaA+Chr) and Ant/(Ant+Phe), like in Figure 7b.

The figure was changed.

While discussing of a series of priority-listed PAH distribution in bottom sediments, in water column, it is necessary to take into account the constants logKow, according to which, hydrophilic PAH will be concentrated in water column, and hydrophobic, high molecular ones – in bottom sediments, the ratio of the latter ones will depend on a pollutants source.

The discussion was expanded.

A very important problem is considered in the clause 3.5, the sediment degradation potential, but the authors did not highlight the following matters. Did a microbial community vary on the sites characterized by the increase of PAH sources intensity compared to former years data, if such ones are available, or to data for relatively clean areas? Which was a fixed response of the ecosystem to the increase of natural objects pollution and which is the own potential of sediments for self-cleaning? I think that it would be interesting for the authors to consider an approach to a study of this problem in

«Anaerobic oxidation of petroleum hydrocarbons ...», Pavlova et al., 2021, Microbial Ecology. https://doi.org/10.1007/s00248-021-01802-y

Unfortunately, there was no previous research into microbial communities capable of degrading PAHs in the water environment of our study area therefore it is impossible to describe the temporary changes in the microbial community structure since pollution increased. We made minor clarification relating to the bacterial composition in the areas next to the study area at the present time.

The approach based on studying the sediment degradation potential under various conditions in the laboratory proposed by the reviewer seems interesting and requires a separate survey, which will probably be carried out in the future.

Minute remarks:

I propose to the authors to change the Figure 2, on which the PAH content levels are presented by circles of different sizes and colors, they present the information unconvincingly. Evidently, it is necessary to present PAH content levels with different forms – circles, triangles, squares and to choose more contrast colors. It would be very useful to mark the depth of sampling.

Corrected

The authors did not indicate the accuracy of PAH determination, the measurements results (a number of significant figures) do not meet the value of confidence intervals in Table S1 and in the text.

Corrected

Authors did not indicate measurement units for PAH concentration for limits of detection, quantification and laboratory blanks.

The information was added

Reviewer 3 Report

Comments and Suggestions for Authors

This manuscript introduced the study on polycyclic aromatic hydrocarbons (PAHs) in the sediment of the Tatar Trough, and the results revealed the ubiquity of PAHs in the deep sea sediments. It is interesting that their distribution showed some difference from many previous studies on sedimental PAHs along coastal areas, e.g., higher concentrations was observed in the deeper sites. The authors accordingly provided reasonable explanation about the their occurrence and source by the PAH composition, isomer ratio and PCA. I think this study can provide important information about persistent organic pollutants in the Tatar Trough, especially the deep sea environment. I recommend it for publication after a mandatory revision.

1.     Generally 16 USEPA priority-listed PAH were mentioned in many previous studies, why were only 14 compounds defined in the present work?

2.     What about the LOD levels in the present study? The result should be provided.

3.     Section 3.2, a positive moderate correlation was obtained between the total PAH concentration and sampling depth, which could be reasonably explained by many aspects, e.g., PAHs in sediment of the north Yellow Sea showed an increasing tendency along the distance from the coastline toward the open sea (Wang et al. Science of the Total Environment 737 (2020) 139535), which was due to hydrodynamic effect. I suggest the authors to provide some positively supporting references in the text.

Comments on the Quality of English Language

The English needs certainly improvement, and it is suggested that the authors contact a natively English speaking colleague. E.g., often present tense is wrongly used ("elevated PAH level are likely", “…carbon content decreased indicates that aging of organic matter in the deep water environment”); “elevated PAH level…” should be “elevated PAH levels” (P1, line 15); “the PAHs” should be “PAHs” (P1, line 24); “Because their environmental resistance the PAHs can be” should be “Because of their environmental resistance PAHs can be… ” (P2, line 47), and so on. Moreover, adverbial modifier is widely used in the text leaving it difficult to understand, e.g., P16-17, P20, P26, etc. In general, more effort should be put in to formulate sentences to make it easier for the reader to follow.

Author Response

We sincerely thank the reviewer for taking the time to review our manuscript. Your comments helped improve the manuscript, especially the discussion, which became much clearer and more relevant to the results.

Additionally, please pay attention to that the data on PAH concentration was changed. While working on correcting the manuscript, we found that we made a mistake when converting data in solution to dry weight. As a result, the PAH concentrations decreased by 2.5 times and thus the discussion of pollution levels was slightly modified. However, this did not affect the results of the relative PAH contribution (compositional profiles), correlation analysis, PCA and isomeric ratio.

This manuscript introduced the study on polycyclic aromatic hydrocarbons (PAHs) in the sediment of the Tatar Trough, and the results revealed the ubiquity of PAHs in the deep sea sediments. It is interesting that their distribution showed some difference from many previous studies on sedimental PAHs along coastal areas, e.g., higher concentrations was observed in the deeper sites. The authors accordingly provided reasonable explanation about the their occurrence and source by the PAH composition, isomer ratio and PCA. I think this study can provide important information about persistent organic pollutants in the Tatar Trough, especially the deep sea environment. I recommend it for publication after a mandatory revision.

  1. Generally 16 USEPA priority-listed PAH were mentioned in many previous studies, why were only 14 compounds defined in the present work?

Acenaphthylene and indenopyrene were not included in the study. We clarified in the relevant section that as acenaphthylene does not fluoresce it wasn’t analyzed. As for indenopyrene, we were limited in purchasing its standard during the experiment.

  1. What about the LOD levels in the present study? The result should be provided.

The information was added.

  1. Section 3.2, a positive moderate correlation was obtained between the total PAH concentration and sampling depth, which could be reasonably explained by many aspects, e.g., PAHs in sediment of the north Yellow Sea showed an increasing tendency along the distance from the coastline toward the open sea (Wang et al. Science of the Total Environment 737 (2020) 139535), which was due to hydrodynamic effect. I suggest the authors to provide some positively supporting references in the text.

The information was added.

Comments on the Quality of English Language

The English needs certainly improvement, and it is suggested that the authors contact a natively English speaking colleague. E.g., often present tense is wrongly used ("elevated PAH level are likely", “…carbon content decreased indicates that aging of organic matter in the deep water environment”); “elevated PAH level…” should be “elevated PAH levels” (P1, line 15); “the PAHs” should be “PAHs” (P1, line 24); “Because their environmental resistance the PAHs can be” should be “Because of their environmental resistance PAHs can be… ” (P2, line 47), and so on. Moreover, adverbial modifier is widely used in the text leaving it difficult to understand, e.g., P16-17, P20, P26, etc. In general, more effort should be put in to formulate sentences to make it easier for the reader to follow.

The phrases were checked.

Round 2

Reviewer 2 Report

Comments and Suggestions for Authors

The manuscript of the article is worthy of publication in the journal Water.